# A Monotone System Generator for Solving Big Data Aggregation Problems

**F.T. Adilova, R.R. Davronov**
Laboratory of Biomedical Informatics
V.I.Romanovskiy Institute of Mathematics, Uzbekistan Academy of Sciences
9 University Street, Tashkent 100174, Uzbekistan
`fatadilova@mathinst.uz, rifqat@gmail.com`

## Abstract

It is well known that the theory of monotone systems transforms clustering from a global optimization problem (which is often NP-hard) into a successive elimination problem solvable in polynomial time. The proposed approach requires minimal a priori information: specifying only the relationship measure of one object with a subset of objects. The algorithm guarantees an exact solution to the stated extremal problem. The approach is based on the concept of a monotone system $< A, F >$, where $A$ is a finite set of objects, and $F(X)$ is an importance (or weight) function defined on subsets $(X \subseteq A)$. The monotonicity condition is: $F(X \setminus \{a\}) > F(X)$. We consider a generator of monotone systems. The proposed procedure for generating a family of monotone systems consists of two stages: i) constructing a set of transformation operators for a monotone system of a sufficiently general form, defined on the same initial set $W$, $|W| = N$; ii) constructing a set of basic functions on the set W. The desired generator is considered as a structure that generates compositional chains of operators over monotone systems selected as basis ones. The proposed extension of the class of basic functions for monotone systems is implemented in a class of Estimate Calculation Algorithms (ECA). A problem statement is formulated for defining a set of three types of basic functions in a monotone system in a class of estimator algorithms. Changing the sets of operators in the basic systems generates a family of monotone systems, which has a wide range of applications, for example, in genetic network analysis, natural language processing (NLP), image processing, and, in general, as a new tool for solving complex problems of structuring large data sets.

## 1 Introduction

The problem of cluster analysis lacks a clearly formalized criterion—all formalizations are subjective and not tied to model constructions. Therefore, fundamental decisions are called into question: the choice of features and their measurement scales, the selection of a measure of similarity between objects, and the interpretation of results. Generally, the clustering problem is formulated as a combinatorial extremization problem. In such problems, there are no effective procedures that would deliver a global extremum to the corresponding functional. Therefore, one can only hope to achieve a local extremum. These specificities are very strict restrictions for applied scientists who use broader concepts in their qualitative descriptions.

To overcome the limitations of known methods, increasingly complex combined analysis processes or specialized methods focused on specific types of information are being used to find precise solutions to the problem. Interesting results in solving this problem were once obtained using the theory of monotone systems, which represents a universal mathematical framework that allows for the removal of these restrictions at the level of problem formulation in general.

Here, clustering is viewed not as a partition of space, but as a process of sequentially identifying "cores" based on certain connectivity functions. It is based on the concept of a monotone system $\langle A, F \rangle$, where $A$ is a finite set of objects, and $F(X)$ is the significance (or weight) function defined on subsets $(X \subseteq A)$. The monotonicity condition is: $F(X \setminus \{a\}) > F(X)$. The motivation

for this study was the desire to develop a method that is sufficiently rich in terms of covering the required number of objects under study, while at the same time reducing the complexity of finding the function. We see the solution as the construction of an ordered sequence of subclasses, in which each subsequent subclass contains functions with a more stringent assessment of the similarity of an element to a subset. A well-known method of this type is the method of Estimate Calculation Algorithms (ECA), which belongs to the school of Academician Yu. I. Zhuravlev (Zhuravlev et al., 1974).

## 2 Relevant Works

The increasingly recognized problem of the current stage of cluster analysis development has generated many accessible programs implementing different methods that generate different cluster structures using the same data. Which structure should be adopted as the final solution? In the context of discrete data analysis (the school of I. Muchnik and J. Mullat), monotone systems provide a unique clustering tool that is fundamentally different from classical methods like $k$-means.

To solve clustering problems, Mullat (1976) proposed an approach called a monotone cluster. The approach is based on assessing the relationships between objects and sets of objects (Kuznetsov & Muchnik, 1982; Aaremaa, 1986). The idea behind this approach is to use a monotonic similarity function between all objects and subsets as input. In a typical situation, such a function can easily be determined based on data of any nature, including digitized text or images.

The functions $F(X)$ can be maximized in a "greedy" manner, selecting the best candidate at each step, which leads to an "optimal" sequence of objects that determines not only the "core"—the set $S$ that maximizes $F(S)$ as a fragment of this sequence—but also the set of its "shells"—the enclosing fragments. The weakest-link functions are only those that satisfy the so-called quasiconvexity condition for all $S, T \subset I$ associated with order structures such as finite "convex geometries" (Mirkin & Muchnik, 2002).

The basic idea is to define a connectivity function that evaluates the "contribution" or "connection strength" of an element within a set. A system is called monotone if, as the set expands, the "connectedness" of an element within it does not increase (or does not decrease, depending on the type of system). The theory proves that for any such system, there exists a unique sequence of nested subsets, each of which is maximally connected for its level. Unlike many NP-hard clustering problems, core search in a monotone system is performed in polynomial time using a simple greedy algorithm (sequentially removing the "weakest" element).

Mirkin (2012) proposed an original theoretical approach that views clustering as a process of reconstructing the hidden structure of data. This approach allows for the unification of disparate methods, such as $K$-means and Ward's hierarchical clustering, into a unified mathematical framework. In (Mullat, 2004) a monotone system is considered as a set of subsets, the elements of which possess certain "indicators" or "credentials". These indicators change monotonically when the composition of the subsets changes. A procedure for identifying the most significant subsystems—cores—is defined. Positive and negative cores represent elements that are most sensitive to changes ("actions") within the system. Removing such a core entails a radical change in the properties of the entire structure.

Muchnik & Shvartser (2010) focus on discrete dynamical systems where state transitions occur stepwise, which is critical for problems of classification, pattern recognition, and data analysis. The methods described in the paper find application in combinatorial data analysis (CDA), bioinformatics (e.g., for gene clustering), and social choice theory. The authors demonstrate that using local properties allows one to construct more efficient algorithms for finding extremal subsystems and kernels of monotone systems, which simplifies the processing of large data sets.

In (Mullat & Leetma, 2016) the authors present algorithms for reordering data tables in order to give them a more informative form. The main goal of the algorithms is to summarize the entire data table using one "average" object, called the "best decision". This object is defined as the one that gives the maximum value of the weight function. All the described methods are based on the mathematical apparatus of monotone systems, which allows one to analyze the dynamics of indicators that increase or decrease in accordance with a partial order in subsets.

Mullat (2024) present an updated version of fundamental research on the theory of extremal subsystems. The revised version focuses on the formulation and proof of a duality theorem, which describes a systematic approach to restricting the search space for kernels and stable sets in monotonic systems. A constructive algorithm for finding extremal subsystems, implemented as a dual scheme, is presented. The methodology is used to analyze data structures, graphs, and networks, allowing one to extract highly cohesive or self-contained units within complex systems.

In his study Mirkin (2007) adapts the rigorous mathematical framework of monotonic systems to solve applied problems in information systems and management. This publication is significant in that it transfers the theoretical developments of the 1970s and 1980s into the context of modern data mining, making them accessible to business analytics professionals.

The work of Yao & Liu (2024) finds the most densely connected groups of users, ignoring "noise" and the periphery (community detection). In the search for groups of genes that behave similarly under certain conditions, monotonic systems make it possible to find stable gene clusters even in the presence of strong noise in the sequencing data (Datta & Datta, 2006). Parupudi (2025) implement feature selection to identify the most informative variables that maintain a monotonic relationship with the target variable using the framework of monotonic systems.

Makdesi et al. (2023) propose a model mapping that provably contains all monotonic functions consistent with the data. This mapping is also minimal in the sense that any set-valued mapping containing all consistent monotonic functions will also include the proposed one. It is shown that this minimal model mapping is interval-valued and allows for a simple construction on a finite partition induced by the data. Since the complexity of the partition increases with the data volume, the authors consider the problem of computing minimal interval-valued model mappings defined on a priori fixed partitions. Finally, algorithms are proposed for constructing guaranteed approximations of unknown monotonic functions based on a discrete data set.

An interesting approach to constructing barrier certificates for unknown monotone systems using a finite number of system trajectories is presented, while providing formal correctness guarantees (Galarza-Jimenez et al., 2025). Given several system trajectories, a family of monotone basis functions is constructed that remain nonincreasing on all trajectories. Using this class of basic functions, sample-based optimization is performed to construct barrier certificates, ensuring system security without the need for additional modeled data or the assumption of Lipschitz continuity.

Thus, the analysis of publications shows that the theory of monotone systems, the foundations of which were formulated by the school of I. Mullat, I. Muchnik and B. Mirkin, presents great potential for its development, which is the subject of this theoretical study. The purpose of this article, taking into account the above-described properties of monotone systems, is to formulate the statement of the problem of specifying in a monotone system a set of basic functions of three types in the class of algorithms for calculating estimates (ECA) (Zhuravlev et al., 1974).

## 3 METHODOLOGY

The theory of monotone systems (MS) identifies a new class of effectively solvable combinatorial-extremal problems and, in this sense, has a broader significance than simply another theory of methods for processing empirical information. An interesting application of the theory of monotone systems is its application to structural methods of data analysis. One of the main elements of this analysis is the identification of sub-matrices that formalize the original structure in the form of an "object-feature" matrix (Braverman & Muchnik, 1983). Such a formalization requires the construction of a flexible mathematical framework with a set of transition operators for moving between different formalizations. This will allow one to "select" the appropriate formal procedure for extracting submatrices and construct a detailed description of the system. This is the idea behind the monotone system generator, as a system for generating a family of monotone systems. In general, the MS generator is represented by a two-stage implementation. In the first, a set of operators for transforming monotone systems of a fairly general form is constructed, and in the second stage, a set of basic functions is designed.

## 3.1 MONOTONIC SYSTEMS

Below, we briefly outline the key principles of monotonic systems theory as presented in Mullat (1977). Let $W$ be a finite set of elements, $|W| = N$, and let $\pi$ be a scalar function which assigns to each pair $(i, H)$, where $H \subseteq W$ is an arbitrary subset of $W$ and $i \in H$, a number $\pi(i, H)$, where the meaning of $\pi(i, H)$ is the affinity of the element $i$ to the subset $H$. $\pi(i, H)$ can also be interpreted as the importance of element $i$ in subset $H$.

For example, if a matrix $\|\rho_{ij}\|_N^N$ of distances between all pairs of elements is given for a set $W$, then the simplest function $\pi(i, H)$ is the sum of the distances from an element to all other elements:

$$\pi = \sum_{j \in H} \rho_{ij}. \tag{1}$$

Then the system $\langle W, \pi \rangle$, consisting of a finite set $W$ with a set of pairs $(i, H)$ given on it and the function $\pi(i, H)$, is called *monotone* if:

$$\pi(i, H \setminus j) \leq \pi(i, H), \quad \forall i, j \in H, \ i \neq j, \ \forall H \subseteq W, \tag{2}$$

or

$$\pi(i, H \setminus j) \geq \pi(i, H), \quad \forall i, j \in H, \ i \neq j, \ \forall H \subseteq W. \tag{3}$$

If inequality (2) is satisfied, the system is called $(-)$-monotone and is denoted by $(W, \pi^-)$, and if condition (3) is satisfied, then $(+)$-monotone and is denoted by $(W, \pi^+)$.

The main goal of MS theory is to identify some extremal subsystem of it, which is called the largest core. On the set $W$ of all subsets of the $(-)$-monotone system, a scalar function $F^-(H)$ is defined, which is assigned to each subset $H \subseteq W$:

$$F^-(H) = \min_{i \in H} \pi^-(i, H), \quad \forall H \subseteq W. \tag{4}$$

**Definition 1.** The *kernels* of a $(-)$-monotone system $\langle W, \pi \rangle$ are those subsets of the set $W$ on which the maximum of the function $F^-(H)$ is achieved.

$$F^+(H) = \max_{i \in H} \pi^+(i, H), \quad \forall H \subseteq W. \tag{5}$$

Let the elements of the set $W$ be ordered arbitrarily. The resulting sequence of elements $A = \{\alpha_1, \alpha_2, \ldots, \alpha_N\}$, where $W = \{\alpha_1, \alpha_2, \ldots, \alpha_N\}$, one-to-one corresponds to the sequence $\bar{H}(A)$, or $\bar{H} = \langle H_1, \ldots, H_N \rangle$ of nested subsets of the set $W$, where $H_1 = W$; $H_2 = H_1 \setminus \alpha_1$; $\ldots$; $H_{k+1} = H_k \setminus \alpha_k$; $\ldots$; $H_N = \{\alpha_N\}$.

**Definition 2.** An ordered sequence $A$ of elements of a set $W$ is called a *defining sequence* $(-)$ of a monotone system $\langle W, \pi \rangle$ if in the corresponding sequence of sets $\bar{H}$ there exists a sequence $\bar{\Gamma} = \langle \Gamma_1, \ldots, \Gamma_P \rangle$, where $\Gamma_1 = H_1 = W$, such that

$$\pi(\alpha_k, H_k) < F(\Gamma_{j+1}), \quad \forall \alpha_k \in \Gamma_j \setminus \Gamma_{j+1}, \ j = \overline{1, p-1},$$
$$F(L) \leq F(\Gamma_P), \quad \forall L \subset \Gamma_P. \tag{6}$$

**Definition 3.** A set $G \subseteq W$ is called a *definable set* of a monotone system $\langle W, \pi \rangle$ if there exists a defining sequence such that $\Gamma_P = G$.

Central theorems play an important role in the theory of MS. The first of these asserts the attainability of the global maximum of the function $F^-(H)$ on the definable set of a $(-)$-monotone system and the fact that all kernels of the $(-)$-MS lie within a defined set. The second theorem establishes the closure of the system of all sets $X$ on $W$ with respect to the binary operation of union of the sets on which the function $F^-$ attains the global maximum.

Thus, the core of the MS is the last set $G = \Gamma_P$ in the sequence $\bar{\Gamma}$ obtained by constructing the defining sequence. Interest in monotonic systems is currently growing due to the problem of Big Data and the increasing demands on the interpretation of the results of structural data analysis. Monotonic systems, as a tool for associative-structural analysis, represent an attempt to formalize analogous processes occurring in the human brain.

## 3.2  ASSIGNING SIMILARITY IN ECA

Estimating the similarity of object parts is the central element of the ECA method. A subset of features $\omega$ ($\omega \subseteq P$) is selected from the set of features. In the feature-object matrix, the set $S'$ of object coordinates corresponding to the subset $\omega$ is called the $\omega$-part of the object. A binary indistinguishability relation $R_\omega(S, S')$ is defined on the set of $\omega$-parts of all objects under consideration. The target similarity function $r_\omega(S, S')$ between $S$ and $S'$ is given by:

$$r_\omega(S, S') = \begin{cases} 1, & \text{if there is a relation } R_\omega(S, S') \text{ between } \omega\text{-parts } S \text{ and } S', \\ 0, & \text{otherwise.} \end{cases} \tag{7}$$

A family of feature subsets $\Omega$ is identified, and a relation $R_\omega(S, S')$ is defined on each element $\omega \in \Omega$. Then, each pair of objects $S$ and $S'$ divides the family $\Omega$ into two disjoint subfamilies $\Omega^0(S, S')$ and $\Omega'(S, S')$:

$$\begin{aligned} &\text{a) } \omega \in \Omega^0(S, S'), \text{ if } r_\omega(S, S') = 0, \\ &\text{b) } \omega \in \Omega'(S, S'), \text{ if } r_\omega(S, S') = 1, \end{aligned} \tag{8}$$

so that $\Omega = \Omega^0(S, S') \cup \Omega'(S, S')$.

As a function of the similarity of a pair of objects in the ECA, we select $|\Omega'(S, S')|$, which is equal to the number of $r_\omega(S, S') = 1$ on the pair $(S, S')$, written in the form:

$$f(S, S') = \sum_{\omega \in \Omega} r_\omega(S, S'). \tag{9}$$

From (8), the flexibility of the ECA becomes apparent, which is ensured by the wide possibilities of changing the family $\Omega$ and relations $R_\omega(S, S')$ and the ability to include a priori information about the analyzed objects in the selection of the family $\Omega$ and in the definition of $R_\omega(S, S')$ on it.

In the case of binary features, the set $\Omega_n$ of all single features is written as:

$$\Omega_n = P_1 : |\Omega_n| = |P| = n.$$

If $\Omega_q$ is the set of all possible subsets of $P$, then

$$r_\omega(S, S') = \begin{cases} 1, & \text{if } \omega\text{-parts } S \text{ and } S' \text{ coincide}, \\ 0, & \text{otherwise.} \end{cases}$$

The similarity functions in the case of binary features $f_1$ and in the case $\Omega_q$ will have the form:

$$f_1(S, S') = n - \rho(S, S'), \tag{10}$$

$$f_2(S, S') = 2^{n - \rho(S, S')}, \tag{11}$$

where $\rho(S, S')$ is the Hamming distance between $S$ and $S'$.

Between a linear function $f_1$ and a nonlinear function $f_2$, there is a whole range of functions of intermediate complexity. As a family of support sets $\Omega$, all subsets whose cardinality does not exceed a certain given value $k$ ($k \leq n$) can be chosen:

$$\Omega_{np} = \bigcup_{i=1}^{k} \Omega_i, \tag{12}$$

where $\Omega_i$ is the family of all subsets of features of power $i$, and $r_\omega(S, S')$ for each $\omega \in \Omega_{np}$ is calculated according to (7). Then, the similarity function $f_{np}(S, S')$ takes the form:

$$f_{np}(S, S') = \sum_{i=1}^{\tilde{k}} C_{n - \rho_i(S, S')}^{i}, \tag{13}$$

where $\tilde{k}$ is determined from the equation:

$$\tilde{k} = \begin{cases} k, & \text{if } k \leq n - \rho(S, S'), \\ n - \rho, & \text{if } k \geq n - \rho(S, S'). \end{cases}$$

The function $f_{np}(S, S')$ is not complex, as it does not require explicit enumeration of the family $\Omega$ of support sets.

Conditions (7) can be relaxed by introducing a threshold $\varepsilon_\omega$:

$$r_\omega(S, S') = \begin{cases} 1, & \text{if } \rho_\omega(S, S') < \varepsilon_\omega, \\ 0, & \text{otherwise.} \end{cases}$$

It is clear that the introduction of $\varepsilon_\omega$ gives a free parameter of the function $f_{np}(S, S')$.

## 3.3 DEFINING THE BASE FUNCTIONS OF MONOTONE SYSTEMS IN THE ECA CLASS

We will consider three types of functions $\pi(i, H)$. The *first type* is a weight function of pairwise connections between an element $i \in H$ and each element of this set:

$$\pi(i, H) = \sum_{k \in H} a_{ik} = \sum_{k \in H} \sum_{\omega \in \Omega} r_\omega(i, k), \quad i \in H, \ H \subseteq W, \tag{14}$$

where $r_\omega(i, k)$ is the similarity function of elements $i$ and $k$.

The *second type* of weighting function differs from the first in that it introduces a "pattern" on the set $H$, which can be selected from $H$ in some way. For example, such a pattern could be some object $i$ for which the majority of objects in this subset "vote", the cardinality of which $|H| = N_H$:

$$\Gamma_H(i) = \max_{j \in 1, N_H} \{\Gamma_1(i), \Gamma_2(i), \ldots, \Gamma_j(i)\}. \tag{15}$$

The number of votes $\Gamma_j(i)$ is determined on the set of $\omega$-parts of all objects under consideration:

$$\Gamma_j(i) = \frac{1}{|H| - 1} \sum_{i,j \in H} f(i, j), \tag{16}$$

where $f(i, j) = \sum_{\omega \in \Omega} r_\omega(i, j)$ is the affinity function of $i$ and $j$.

Interestingly, a special "object" or "anti-pattern" can be used as a representative of a set $H$. Formally, the problem of identifying special objects is represented as a partial approximation problem. Then, a representative of a set $H$ can serve as a pattern or anti-pattern for the weight functions of the second-type MS.

Weight functions of the *third type* measure the relationship between an element and a subset as the sum of the relationships between the element and the group for each individual feature. Each feature $j$ is characterized by a certain weight $\omega_j(H)$, which depends on the composition of the set $H$ ($j$-th feature weight). Obviously, $\omega_j(H)$ must be monotonic:

$$\omega_j(E) \le \omega_j(H), \quad \forall E \subseteq H \subseteq W, \ \forall j \in Y.$$

In general, the measure of importance of the $l$-th feature $P(l)$, which is understood as its ability to "separate" an object from a certain set, is calculated as:

$$P(l) = \frac{\sum\limits_{t=1}^{m-1} \sum\limits_{j=t+1}^{m} C_{r(S_t, S_j)}^k - \sum\limits_{t=1}^{m-1} \sum\limits_{j=t+1}^{m} C_{r(S_t, S_j) \setminus l}^k}{\sum\limits_{t=1}^{m-1} \sum\limits_{j=t+1}^{m} C_{r(S_t, S_j)}^k}, \tag{17}$$

where $C_{r(S_t, S_j)}^k$ is the measure of the similarity of a pair of rows $S_t$, $S_j$ on a supporting subset of the cardinality $k$ in the case of the deletion of the $l$-th feature. The cardinality of the subset $k$ is an important parameter, and its optimization methods are the subject of future research in the context of MS.

The measure of the importance of the $l$-th feature of the object $S_t \in H$:

$$P(l)_{S_t} = \frac{\sum\limits_{j=1}^{|H|} C_r(S_t, S_j) - \sum\limits_{j=1}^{|H|} C_r(S_t, S_j) \setminus l}{\sum\limits_{j=1}^{|H|} C_r(S_t, S_j)}, \quad j, t = \overline{1, |H|}, \ j \ne t. \tag{18}$$

Then the relation function $\pi(S_t, H)$ of the element $S_t$ and the subset $H$:

$$\pi(S_t, H) = \sum_{l=1}^{n} P(l)_{S_t} \tag{19}$$

shows the distance of $S_t$ from the subset $H$ by all features, weighted in accordance with (18).

There is a known option for determining the weight of a feature in the process of constructing a defining sequence of a monotonic system, which leads to the standard form of a linear programming problem with respect to an unknown system of weights (Eshonkulov, 1993). This study develops the classical direction of Yu. I. Zhuravlev's pattern recognition, adapting it to the tasks of taxonomy (clusterization). A distinctive feature is adaptive methods for adjusting algorithm parameters directly to a specific data sample, which improves the accuracy of classification and identification of significant features.

## CONCLUSION

Defining a system of base functions for monotonic systems within the class of Estimate Calculation Algorithms (ECA) provides a significant advantage in constructing efficient procedures for identifying the core of a monotonic system, as well as in enhancing the interpretability of the resulting solutions. The presented formalization of constructing base functions within the ECA framework remains insufficiently explored and is, therefore, a subject for future research, including experimental and comparative studies of the proposed approach.

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
