# OpenReview forum: "A Monotone System Generator for Solving Big Data Aggregation Problems"
_mathai.club/MathAI/2026/Conference — 2026 Oral_

### Official Review · Reviewer_sjGc · 2026-03-13
**The research analyzes problems of monotone systems through basic evaluation algorithm functions, proposing a formalized model of subclass sequences for data analysis and building on fundamental works by authoritative scholars with detailed mathematical calculations. While the approach demonstrates clarity, originality in expanding basic functions, and significance for experiments—highlighting strengths in model descriptions—the main drawback is the lack of practical examples extrapolating to data extraction methods.**

**Rating:** 6
**Confidence:** 3

**Review:**

The research examines problems of monotone systems through the lens of basic evaluation algorithm functions embedded in the system. It addresses the lack of substantiated formalization of fundamental solutions via cluster analysis. A formalized model of the sequence of subclasses in a monotone system is proposed, applied to data analysis.

Quality: the paper presents a mathematical model of a monotone systems generator, algorithmic calculations of evaluations, and detailed characterization of the components of these models. The study is grounded in fundamental scientific works by authoritative scholars. Definitions and principles underlying the theory of monotone systems are provided.

Clarity: the provided definitions and components of monotone systems theory are explained in light of the scientific principles established by the theory's founders. A characterization of the components used in mathematical calculations is given.

Originality: the application of fundamental theoretical principles of monotone systems theory to modern information extraction methods is noted. Attempts are made to expand the class of basic functions in monotone systems through the formulation of types of basic functions and their variations.

Significance: the proposed approach forms a scientific-theoretical foundation for experimental research in testing basic functions during the calculation of evaluations in monotone systems.

Pros: the paper presents existing fundamental models of monotone systems. An approach to defining basic functions, their role in monotone systems, and their application to modern data extraction tools is proposed.

Cons: the need to extrapolate the proposed approach to practical examples of existing data extraction methods is evident.

---

### Official Review · Reviewer_cUHP · 2026-03-13
**Good mathematical paper without concrete connection with practical examples**

**Rating:** 7
**Confidence:** 2

**Review:**

I am not specialist in Monotone system, but article looks like good mathematical work.

---

### Official Review · Reviewer_sppD · 2026-03-13
**The paper identifies a meaningful problem at the intersection of two established fields and deserves to be presented and discussed at MathAI 2026, even though the amount of novel findings is small.**

**Rating:** 6
**Confidence:** 1

**Review:**

This paper aims to bridge two established mathematical frameworks for data analysis: the theory of Monotone Systems (MS), primarily developed by Mullat, Muchnik, and Mirkin, and the class of Estimate Calculation Algorithms (ECA) from the school of Acad. Zhuravlev. The authors provide a literature review of monotone systems and their applications, outline the fundamental principles of MS theory, and describe the mechanism for defining similarity functions within ECA. The core contribution is stated as the generator of monotone systems which consists of two stages: i) constructing a set of transformation operators for a monotone system of a sufficiently general form, defined on the same initial set W; ii) constructing a set of basic functions on the set W. The desired generator is considered as a structure that generates compositional chains of operators over monotone systems selected as basis ones. The proposed extension of the class of basic functions for monotone systems is implemented in a class of ECA. The three proposed function types are: 1) a sum of pairwise similarities, 2) a function based on a "pattern" or "vote" within the set, and 3) a function incorporating feature weights that depend on the set's composition.

Major remark

The primary contribution, the definition of the three base functions, is presented in a vague, non-rigorous manner. I hope the author will improve this, if the paper is accepted.

Minor remarks

1. There is no proof that the proposed functions satisfy the fundamental monotonicity conditions (inequalities (2) or (3)).
2. The "Monotone System Generator" mentioned in the title and abstract is never clearly defined. The reader is left to infer that it is the two-stage process described in the Methodology, but the connection between the "set of operators" and the "set of base functions" is not elaborated.
3. The notation becomes sloppy in Section 3.3. For example, in describing Type 3 functions, the subscript notation is inconsistent and confusing (e.g., `S_{\bar{j}}` in equation (18) is not defined).
4. The connection of the reviewed works to the specific proposed generator is not always clear. For instance, the detailed discussion of barrier certificates (Galarza-Jimenez et al., 2025) feels tangential.
5. The paper lacks a concrete example or a small case study. Even a simple toy example with a handful of objects and binary features, demonstrating how one of the proposed `π(i,H)` functions would be calculated and whether it yields a monotone system, would dramatically improve the paper's clarity and impact.
6. The word "transforms" in the first line of the abstract is irrelevant. Usually, the term "transformation", applied to optimization  problems, implies a reduction, allowing to obtain exact solution. If this is implied here and if the transformation is computable in polynomial time, then the authors claim that they've solved the P=NP problem. I guess, that they mean something different here.

In general, I think that the paper identifies a meaningful problem at the intersection of two established fields and deserves to be presented and discussed at MathAI 2026, even though the amount of novel findings is small.

---

### Decision · Program_Chairs · 2026-03-14

**Decision:**

Accept (Oral)

**Comment:**

Dear Author(s),

On behalf of the Program Committee of the International Conference on Mathematics of Artificial Intelligence (MathAI 2026), we are pleased to inform you that your paper has been accepted for an oral presentation at MathAI 2026.

Your paper was evaluated through a rigorous two-stage review process involving both automated screening and expert review by members of the Program Committee. The reviewers recognized the quality and contribution of your work.

Presentation details:

- Format: Oral presentation (15–20 minutes + 5 minutes Q&A)
- Mode: You may present either in person (offline) at the conference venue in Sirius, Russia, or remotely via Zoom. Please indicate your preferred mode when confirming your participation.
- Conference dates: Marh 30 - April 3, 2026
- Website: https://mathai.club

Next steps:

1. Please confirm your participation and presentation mode by replying to this email mathai.club@yandex.ru no later than March 15, 2026 18:00 Moscow time.
2. If you plan to attend in person, the organizing committee will provide accommodation details separately.
3. Please prepare your final camera-ready manuscript according to the formatting guidelines available at https://mathai.club and upload it to OpenReview by March 15, 2026 18:00 Moscow time.

Should you have any questions regarding the program, logistics, or your presentation slot, please do not hesitate to contact us.

We look forward to your contribution to MathAI 2026.

With kind regards,

MathAI 2026 Program Committee
International Conference on Mathematics of Artificial Intelligence
https://mathai.club
OpenReview: https://openreview.net/group?id=mathai.club/MathAI/2026/Conference
Telegram: https://t.me/MathAI_club
Email: mathai.club@yandex.ru